# Modified Hexagonal Split Ring Resonator Based on an Epsilon-Negative Metamaterial for Triple-Band Satellite Communication

**DOI:** 10.3390/mi12080878

**Published:** 2021-07-26

**Authors:** Salah Uddin Afsar, Mohammad Rashed Iqbal Faruque, Mohammad Jakir Hossain, Mayeen Uddin Khandaker, Hamid Osman, Sultan Alamri

**Affiliations:** 1Space Science Centre (ANGKASA), Institute of Climate Change (IPI), Universiti Kebangsaan Malaysia, Bangi 43600, Malaysia; P108627@siswa.ukm.edu.my; 2Department of Electrical and Electronic Engineering, Dhaka University of Engineering & Technology (DUET), Gazipur 1707, Bangladesh; mohd.jakirhossain@gmail.com; 3Centre for Applied Physics and Radiation Technologies, School of Engineering and Technology, Sunway University, Bandar Sunway 47500, Malaysia; mayeenk@sunway.edu.my; 4Department of Radiological Sciences, College of Applied Medical Sciences, Taif University, Taif 21944, Saudi Arabia; ha.osman@tu.edu.sa (H.O.); s.alamri@tu.edu.sa (S.A.)

**Keywords:** triple band, SSR, ε-negative material, satellite communication

## Abstract

A triple-band epsilon-negative (ENG) metamaterial based on a split ring resonator (SSR) with a modified hexagonal-shaped metal strip proposed in this study is a new combination of a single slit square resonator and a modified hexagonal-shaped metal strip. The desired unit cell FR-4 (lossy) that was selected as the substrate was 1.6 mm thick. Following the assessment of the unit cell, a high-frequency electromagnetic simulator like the computer simulation technology (CST) microwave studio was applied to assess the S-parameters. The proposed design exhibited resonance at 2.89, 9.42, and 15.16 GHz. The unit cell also demonstrated negative permittivity in the frequency ranges 2.912–3.728 GHz, 9.552–10.144 GHz, and 15.216–17.328 GHz, along with a negative refractive index. An effective medium ratio (EMR) of 11.53 is an indicator of the goodness of the metamaterial unit cell. It is deliberate at the lowermost resonance frequency of 2.89 GHz. Moreover, the simulated results that were validated using HFSS and equivalent circuit model indicated slight variations. The proposed design was finalised based on several parametric studies, including design optimisation, different unit cell sizes, various substrate materials, and different electromagnetic (EM) field propagations. The proposed triple band (S, X, and Ku bands) negative permittivity metamaterial unit cell can be utilised for various wireless applications, such as microwave communication, satellite communication, and long-distance radio communication.

## 1. Introduction

Materials found in natural surroundings can achieve positive permittivity and permeability. Contrarily, metamaterials yield both negative permittivity and negative permeability or either trait can be negative by itself. An engineered structure indicating both negative permittivity and permeability is called a double negative (DNG) metamaterial, while a structure manifesting either negative value is called a single negative (SNG) metamaterial [1,2]. DNG and SNG metamaterials are extensively used in different technological developments. The use of metamaterials in synthetic dielectrics of microwave application began just after the Second World War. The advancements in material sciences led to the industrialisation of metamaterials in radio frequency, microwave, and optical appliances. Meanwhile, the introduction of metamaterials with a negative refractive index was utilised to shield an object from view, termed cloaking [3]. Metamaterials are usually prepared by repeating the unit cell or proposed pattern, taking into consideration the wavelength of the desired phenomenon. Moreover, the properties of metamaterials depend on their design, size, forms, or assembly instead of the material ingredients [4]. In short, uncommon precise orientation, shapes, or sizes determine the unique or smart novel properties of metamaterials, particularly EM properties. Material researchers are passionate and interested in this field to harvest the benefits of metamaterials (absorbing, cloaking, blocking, bending or enhancing EM radiation) due to their novel EM properties [5,6].

The extant literature has indicated the uses of metamaterials in many applications. For instance, the L band stretchy metamaterial with an EMR of 4.8 using nickel aluminate substrate was designed for a sensing application [7]. Sekip Dalgac et al. [8] presented an S-shaped metamaterial comprising a rectangular split-ring resonator to forecast moisture in the air and temperature. In 1968, Victor Veselago, a Russian physicist, introduced a theoretical explanation of materials with negative permittivity and negative permeability, called the left-handed (LHM) characteristics [9]. Meanwhile, Pendry et al. revealed an SSR metamaterial based on synthetic magnetic materials with different shapes of SRR (circular, rectangular) on a dielectric substrate [10]. In another study, an S-shaped resonator was bordered by a ground ring [11]. A microstrip line was deployed in this metamaterial for feeding. This sort of metamaterial used an L-band with 1.8 GHz resonance frequency. An amalgamated metamaterial (12 × 12 mm^2^ dimension) with an EMR of 7.44 recorded resonances at S, C, and X bands [12]. Meanwhile, a design of a Greek-key pattern operated at a frequency range from 1 to 5 GHz was fabricated on a substrate of Rogers RT5880 (10 × 10 mm^2^ dimension). The design measured resonances at 2.4 GHz, 3.5 GHz, and 4 GHz [13]. After three decades, Smith et al. [14] described an experimental demonstration of functioning EM metamaterials using straightly mounding, intermittently, SRR, and thin-wire structures. In 2001, Shelby et al. [15] demonstrated a negative refractive index using a two-dimensional array of repeated unit cells that were also a combination of wire strips and split-ring resonators.

On the other hand, Winston E. Kock developed a material that has analogous features to the metamaterials in 1940 [16]. Flame Retardant-4 (FR-4) material is commonly used to design substrates. Modern technologies demand compounds that are cost-effective, compact, and lightweight, which led to the fabrication of low-cost and easily designed substrates. For example, an ENG metamaterial with an EMR value of 3.5 exhibited EM cloaking operation as it covered the C-band [17]. In another study [18], a dual SRR of hexagonal shape with a gap (GCHSRR) metamaterial (10 × 10 mm^2^ dimension) exhibited microwave absorbance uses as it covered the C- and X-bands. In recent times, metamaterials have been aimed at supporting various frequency bands. Each frequency band has widespread use; the S-band covers a frequency range of 2–4 GHz; the X-band covers 8–12 GHz, and the Ku-bands covers 12–18 GHz. The three bands are extensively utilised in satellite communications [19]. Among them, the terrestrial signals can only slightly affect the Ku-band, which operates in a high-frequency range that allows the use of trivial-sized dish antennas for VSAT uses. Meanwhile, a metamaterial with a harmonising SRR (5 × 5 mm^2^) and EMR of 8 demonstrated a resonance frequency of 7.5 GHz [20].

This study proposed a new and novel metamaterial assembly whose unit cell consists of one SRR with a modified hexagonal-shaped metal stripe. It demonstrated resonance frequencies in the S-, X-, and Ku-bands (2–4 GHz, 8–12 GHz, and 12–18 GHz) in the microwave spectra, apart from indicating negative permittivity and negative refractive index at the same frequencies. Furthermore, the reliability of the performance of the proposed metamaterial was tested using different validation tests. Equivalent circuits were also designed in ADS, HFSS design, 2 × 1 and 2 × 2 arrays to simulate validation tests. The analyses were performed using different effective parameters like permittivity, permeability and the refractive index, together with transmission and reflection coefficients. The EMR was also used as a comparison of performances in the result section.

## 2. Design of the Unit Cell and Simulation

A vivid illustration of the projected metamaterial unit cell illustrated in Figure 1a is a combination of a square-shaped single slit resonator (SSR) and a modified hexagonal metal strip. Figure 1b demonstrates the simulation setup of the unit cell. A 0.035 mm thick copper (annealed) was used for SSR and modified hexagonal-shaped strip. Moreover, the dielectric substance FR-4 (lossy) with a thickness of 1.6 mm was used as a substrate with a dielectric constant of 2.2 and a loss tangent of 0.025. The dimension of the square shaped-substrate was 9 × 9 × 1.6 mm^3^. The length and width of the unit cell SSR metal strip were defined as W1 and W2, respectively, while the length and width of the substrate were a and b, respectively. The slit width (g) and height were each 0.2 mm, with a 0.5 mm gap (G) between the two strips of the hexagon. However, the width (w) of all the metal strips in SSR was 0.5 mm, and the width of the hexagonal arm strips (y) was 1 mm. According to the simulation setup of the proposed unit cell (Figure 1b), an EM wave was applied along the z-axis. In addition, a perfect electrical and magnetic boundary were applied consecutively along the x-axis and y-axis. The dimensions of the unit cell, including all parametric values, are listed in Table 1.

## 3. Effective Medium Parameters Extraction Method

The EM simulator CST was applied for the mathematical and geometrical analyses based on a definite integration procedure. This method provided tractability in different composite symmetrical structure modelling and integrated the arbitrary distribution of materials with their properties. Simultaneously, a frequency-domain solver using a tetrahedral mesh in CST was used to simulate the proposed structure. The EM waves were fixed to promulgate to the z-axis. Perfect electric and magnetic fields were applied along the x- and y-axes as a boundary condition to simulate 2.0–18.0 GHz of the resonance frequency. Additionally, transmission coefficient (S21), permittivity (ε_r_), and refractive coefficient (n_r_) were evaluated to determine the EM properties of the proposed structure using the Nicolson–Ross–Weir technique [21].
(1)$$\mathrm{Permitivity},\varepsilon_{r}=\frac{c}{j\pi \mathrm{fd}}\times \frac{\left\{1-\left(S21+S11\right)\right\}}{\left\{1+\left(S21+S11\right)\right\}}$$
(2)$$\mathrm{Permeability},\;\mu_{r}=\frac{c}{j\pi \mathrm{fd}}\times \frac{\left\{1-\left(S21-S11\right)\right\}}{\left\{1+\left(S21-S11\right)\right\}}$$
$$\mathrm{And}\;n_{r}=\sqrt{\varepsilon_{r}\mu_{r}}$$
(3)$$\begin{array}{c} \mathrm{Or},\;n_{r}=\frac{c}{j\pi \mathrm{fd}}\sqrt{\frac{\left\{1-\left(S21+S11\right)\right\}}{\left\{1+\left(S21+S11\right)\right\}\;}\times \frac{\left\{1-\left(S21-S11\right)\right\}}{\left\{1+\left(S21-S11\right)\right\}\;}}\\ =\frac{c}{j\pi \mathrm{fd}}\sqrt{\frac{\;{\left(S21-1\right)}^{2}-S11{}^{2}}{\;{\left(S21+1\right)}^{2}-S11{}^{2}}} \end{array}$$
where, c, f, and d represent the speed of light, frequency, and the thickness of the substrate, respectively. S11 represents the reflection coefficient, and S21 is denoted the transmission coefficient. To obtain the refractive index (n_r_), relative permittivity (ε_r_), and permeability (µ_r_), MATLAB codes were established based on Equations (1)–(3). The results of the parameters extracted through the NRW method was compared with the CST results.

## 4. Parametric Study

### 4.1. Design Procedure

To maximise the benefit from a design, the best design was selected among the available trial designs. An iterative method was performed to assess the response of the unit cell. The length, width, the distance between the SSR, and the size of the slits were changed based on trial-and-error. Figure 2 represents some of the trial designs. Design-1 of Figure 2 contains a single slit square ring with a uniform width of 0.50 mm. This is an SSR with a dimension of 8 × 8 mm^2^ and a thickness of 0.035 mm with a split gap of 0.40 mm. This SSR demonstrated two major responses at 3 GHz and 10 GHz. Design-2 illustrates a hexagonal loop with a width of 1 mm that was added to Design-1 (Figure 2b). Following the simulation, the resonance frequencies recorded were at 3 GHz, 10 GHz, and 17 GHz. Meanwhile, the modification to Design-2 gave rise to Design-3, which had an addition of two metal strips that were 6 mm long and a split gap of 1 mm. This modified design recorded resonances at 3 GHz and 10 GHz. Finally, the placement of two more metal strips with the same split gap in Design-3 in a horizontal orientation on the hexagon arms was represented as the proposed design (Figure 2d). The proposed design measured three major frequencies at 2.89 GHz, 9.42 GHz, and 15.16 GHz. Figure 3 indicates the transmission coefficient (S21) for the four different designs.

The proposed design has a substrate with a dimension of 9 × 9 × 1.6 mm^3^; the length and wide of the SSR unit cell metal strips were W1 = 8 mm and W2 = 8 mm, with a split width (g) of 0.20 mm. Moreover, G, the split gap between the placed metal bar strips of the hexagon was measured at 1 mm with every metal strip at 0.5 mm width. The proposed structure demonstrated a maximum number of resonance (frequency bands) with negative permittivity and negative index. The dimensions of the suggested metamaterial unit cell, including all designs tranmission coefficients results values, are listed in Table 2.

### 4.2. Effects of Changing of the Unit Cell Size

To optimise the size of the proposed metamaterial, the unit cell with different substrate sizes were simulated. Based on Figure 4, the substrate of sizes 11 mm and 13 mm with a thickness of 1.6 mm exhibited three resonances with low bandwidth. Contrarily, the 9 mm substrate with the same thickness demonstrated three major resonances at 2.89 GHz, 9.42 GHz, and 15.16 GHz. Hence, the 9 × 9 × 1.6 mm^3^ dimension was fixed as the proposed unit cell size.

### 4.3. Effects of Changing Substrate Properties

According to a study, two types of Rogers (RT 5880 and RT 6002) and an FR-4 (lossy) with the same dimension (9 × 9 × 1.6 mm^3^) were used to observe the properties of different substrates materials. The loss tangent and paramagnetic constant for substrate RT 5880 were 0.0004 and 2.2, as for RT 6002 were 0.0037 and 3.48, respectively. The S21 depicted in Figure 5 indicated three sharp resonances with narrow bandwidths for RT 5880 and RT 6002. Substrate FR-4 (Dk 2.2 and loss tangent 0.025) demonstrated three resonances with satisfactory bandwidth covering S-, X-, and Ku-bands. A high EMR of 11.53 was calculated for the first resonance frequency (2.89 GHz). Therefore, FR-4 (lossy) was selected as a substrate material for the fabrication of the proposed unit cell.

### 4.4. Effects of Changing the EM Field Propagation Direction

The electric field (Ey) was applied towards the y-axis, while the magnetic field (Hx) was applied towards the x-axis (Figure 6a). The resonances recorded at both axes were 6 GHz and 15 GHz, respectively. If the fields are interchanged with each other, an electric field (Ex) was applied in x-direction and the magnetic field (Hy) in y-direction (Figure 6b); three major resonances were observed at 2.89 GHz, 9.42 GHz, and 15.16 GHz (Figure 7).

## 5. Surface Current, Electric Field, and Magnetic Field Analysis

The EM characteristics of metamaterials rely on different forces and fields, which are depended upon to produce charge. Maxwell’s curl equations are the best tools to explain the generated electric and magnetic fields [22].
(4)$$\nabla \times H=J+\frac{\partial D}{\partial t}$$
(5)$$\nabla \times E=\frac{\partial H}{\partial t}$$
where vector operator, $\nabla =[\frac{\partial x}{\partial t},\frac{\partial y}{\partial t},\frac{\partial z}{,\partial t}]$.

Equations (4) and (5) are not sufficient to describe the relationship between matter, the electric field, and the magnetic field. However, these limitations were solved using the following equations.
(6)$$\varepsilon \;\left(t\right)=\frac{D\left(t\right)}{E\left(t\right)}$$
(7)$$µ\;\left(t\right)=\frac{B\left(t\right)}{H\left(t\right)}$$

The equations presented above were used to assess the surface current, magnetic, and electric fields functions. The current distribution at three different resonance frequencies was 2.89 GHz, 9.42 GHz, and 15.16 GHz (Figure 8). The current flow differs with the change of frequency. Based on Figure 9a–c, the outside square ring of the unit cell was determined to have been strongly dominated by the surface current. In other words, it meant that the outer ring belonged to the resonance at 2.89 GHz. Moreover, it was notable that the hexagonal ring was also subsidised as a major part of the current. The magnetic field distribution for the proposed unit cell at various resonance frequencies (2.89 GHz, 9.42 GHz, and 15.16 GHz) is depicted in Figure 10a–c.

According to the Biot–Savart and Ampere’s law, a magnetic field is produced due to the motion of charged particles or electrical charges [23]. The flow of higher electrical charges increased the strength of the magnetic field that was observable from the H field (Figure 10). The magnetic field centered in a wire at a distance (r) follows the equation (using the Biot–Savart law), $B=\frac{\mu_{0}I}{2\pi r}$, where $\mu_{0}$ is the permeability of free space. A similar outcome was also yielded in the E field (Figure 9). Compared to the electric field pattern presented in Figure 5 with the H field presented in Figure 6, it is evident that fewer changes in the H field at a particular point also measures low electric field intensity. Furthermore, the electric field intensity at the point of splits was higher since the split in the resonator ring gap formed a capacitor.

## 6. Circuit Model of the Projected Metamaterial Unit Cell

The proposed metamaterial involved both inductive and capacitive components. All the metal bars and the metal strips of the rings (both SRR and hexagonal) acted as inductors, while the gaps within the metal bar or rings formed capacitors. Therefore, an LC resonance circuit was formed in this metamaterial unit cell. As such, the resonance frequency (f) can be derived using Equation (8) [17].
(8)$$f=\frac{1}{2\pi {\left(LC\right)}^{\frac{1}{2}}}$$
where *L* is inductance, and *C* is capacitance. The capacitance of the capacitor formed by the slits can be calculated using Equation (9).
(9)$$C=\in_{0}\in_{r}\frac{A}{d}(F)$$
where, $\in_{0}\;\mathrm{and}\;\in_{r}$ represent the permittivity in free space and the relative permittivity, respectively, and A is the cross-sectional area of the conducting strip, while d is the split gap.
(10)$$L\left(nH\right)=2\times 10^{-4}l\left[\ln\left(\frac{l}{w+t}\right)+1.193+0.02235\left(\frac{w+t}{l}\right)\right]K_{g}$$

Here, K_g_ = Correction factor, w = width, l = length, and t = thickness of the strip

Figure 11 represents an estimated equivalent circuit for the designed metamaterial unit cell. At the equivalent circuit, L1, L2, L3, L4, L5, and L6 refer to the inductors, while C1, C2, C3, C4, and C5 are capacitors. The circuit was optimised and simulated using the advanced design system (ADS). As such, inductor L1 and capacitor C1 were responsible for the first resonance (2.89 GHz), where L1 regulated the resonance frequency of S21, with C1 as the biasing factor that modifies the magnitude. Whereas C3 is the coupling capacitor, L3 and L4 refer to two series inductors, while L5 and L6 refer to two parallel inductors that contribute to the other two resonances. Figure 12 illustrates a comparative view of the simulated S21 derived from the CST and ADS describing the closest similarities between these two S21 values. However, the impedance was derived for equivalent circuits based on the Thevenin theorem.

## 7. Results and Discussion

The mathematical analyser CST was employed in this study to determine the S-parameters and different coefficients of the unit cell. Figure 13 represent the reflection coefficient (S11) and S21. The unit cell recorded resonance frequencies at 2.89 GHz, 9.42 GHz, and 15.16 GHz with −25 dB, −22.24 dB, and −28 dB amplitude, respectively. As for S11, negative peaks were observed at 3.78 GHz and 10.20 GHz with −23.72 dB and −8.98 dB amplitude, respectively. Bandwidths of 0.39 GHz at S-band, 0.94 GHz at X-band, and 2.64 GHz at Ku-band are also depicted in Figure 13. Moreover, the numerical data were extracted for permittivity, permeability, and refractive index using MATLAB code (Figure 14, Figure 15 and Figure 16). According to Figure 13, it is found that every resonance of S21 was tracked by an S11 minimum. Since every minimum frequency of S21 was lower than that of the corresponding S11 minimum frequency, every resonance was treated as an electrical resonance in the unit cell. Meanwhile, Figure 14; Figure 15 represent the results of permittivity and permeability. It is observed that the value permittivity goes through from maximum to minimum while resonance takes place in S21. The figures also denoted that minimum values of µ were derived from minimum S11 points. The values of permittivity changed from positive to negative when µ positively changed the quantity. Since the refractive index was related to the frequency, the near-null property at the negative permittivity region is depicted in Figure 16. Due to the negative permittivity of the projected unit cell in a certain frequency region, the proposed design was termed a negative-permittivity (ENG) metamaterial. Applying the Drude function [24].
(11)$$\varepsilon \left(\omega \right)=1-\frac{\omega_{p}^{2}}{\omega \left(\omega +i\Gamma \right)}$$
where Γ is the attenuation function due to energy disintegration in the plasmons, and $\omega_{p}$ is the plasmon frequency which is determined by $\omega_{p}=\frac{dq^{2}}{\epsilon_{0}m}$. Also, d represents effective density, q for electron charge, ∈_0_ for permittivity of free space, and m for the mass of the electron. If Γ ≈ 0 then ω < ω_p_ and then permittivity (ε) will be negative. Thus, the EM wave will not propagate. This unique property (negative permittivity) of the metamaterial can widely be used in various types of communication systems, for instance, bandwidth improvement of the antenna [25] and microwave filter. Meanwhile, the negative refractive index property can be utilised for cloaking, sensing, and enhancing antennas. To validate the authentication of the result, the unit cell was simulated by CST and HFSS (S11 and S21) are illustrated in Figure 17. The extracted parameters and their respective frequencies values are listed in Table 3 and comparison of resonance frequencies between the unit cell and arrays are listed in Table 4.

### Array Structure

The unit cell does not function satisfactorily as an independent unit. Hence, an array of unit cells is the best technique to obtain the expected EM properties from a metamaterial. The 2 × 1 and 2 × 2 arrays of the proposed unit cell with the same substrate material were simulated by CST (Figure 18a,b). A parametric study was performed using 2 × 1 and 2 × 2 arrays to investigate the coupling effect between the designed unit cells for different combinations of the array recorded nearly identical S21 responses to the unit cell. Therefore, the efficiency of the arrays was manifested. Figure 19 demonstrates the graphical presentation of S11 and S22 comparison by CST for array combination and the proposed unit cell. Although the resonance frequencies were a little distorted in the new array metamaterial, the triple band resonance properties were still functioning based on the simulated results of the array presented in Figure 19. The authentication between the previous configurations and proposed configuration values are listed in Table 5.

On the other hand, the symmetric pattern of the used SSR and hexagonal metal strip were important in achieving the abovementioned performance. This symmetric pattern of the unit cell helped to decrease the induction effects of the magnetic field. For this reason, the transmission coefficient (S21) is not affected by the harmonic and shows almost the same scattering behaviour and EM properties in the full frequency spectrum.

## 8. Conclusions

This study discussed a metamaterial with negative permittivity designed on an FR-4 substrate with resonance frequencies at 2.89 GHz (S-band), 9.42 GHz (X-band), and 15.16 GHz (Ku-band). Such metamaterials are nowadays designed and fabricated to influence EM radiation using new techniques. Many vital properties (optical and EM properties) of various natural materials have been converting through this chemistry. To investigate the performance of the proposed design, the dependency on the dielectric constant and an array or combination of the metamaterial characteristics (here three validations for a single unit cell and array) were incorporated to produce a comparative analysis. An extensive investigation was performed to analyse the surface current, electric field, and magnetic field distribution. The DPS-ENG media properties were also assessed using an EM wave interaction. The vital parameters like permittivity, permeability, and refractive index were determined using MATLAB coding, whereby the results exhibited negative permittivity with a negative refractive index. Whereas, 2 × 1 and 2 × 2 array metamaterials that were simulated using the same substrate material demonstrated minimal changes in resonance frequencies. On the other hand, the efficiencies of an equivalent circuit that was designed for the desired unit cell were tested via ADS. This result indicated S21 to be very close to the CST results. The proposed unit cell also yielded a resonance frequency of 2.89 GHz with an EMR of 11.53. Since S-, X-, and Ku-bands are widely utilised in satellite and RADAR communications, these tri-band spectra were utilised for microwave communication, long-distance radio communication, and satellite communication. Although this study utilised a simple and compact metamaterial structure, the proposed design manifested triple-band resonance frequencies with excellent EMR value (11.53). In conclusion, this unique metamaterial structure adapted several validation processes that led to its novelty.

## Figures and Tables

**Figure 1 micromachines-12-00878-f001:**
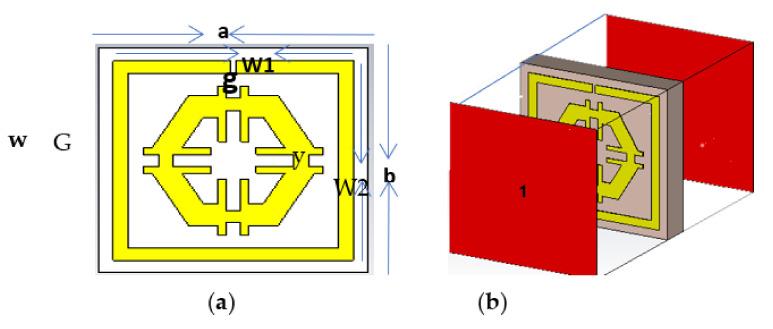
(**a**) Unit cell and (**b**) Simulation arrangement.

**Figure 2 micromachines-12-00878-f002:**
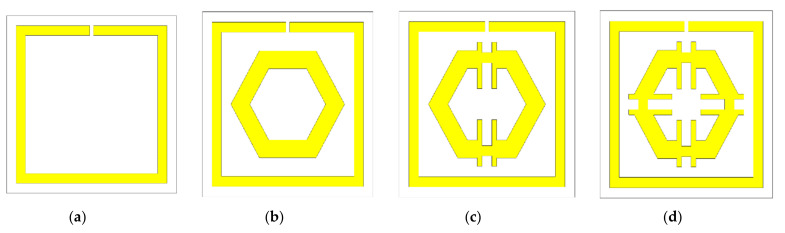
Trial design: (**a**) Design-1, (**b**) Design-2, (**c**) Design-3, (**d**) proposed design.

**Figure 3 micromachines-12-00878-f003:**
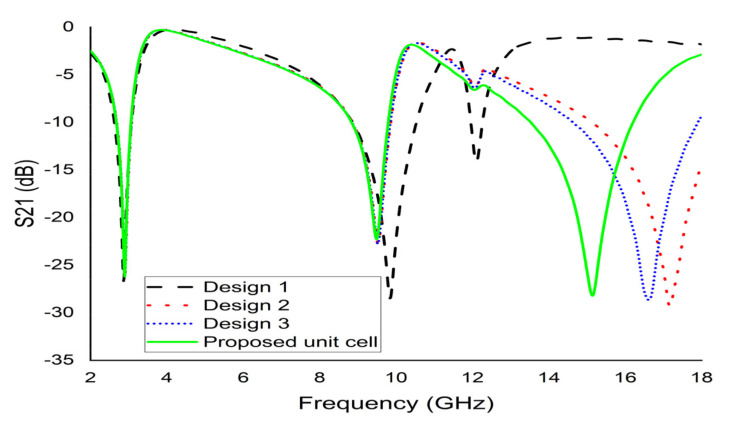
S21 for various designs.

**Figure 4 micromachines-12-00878-f004:**
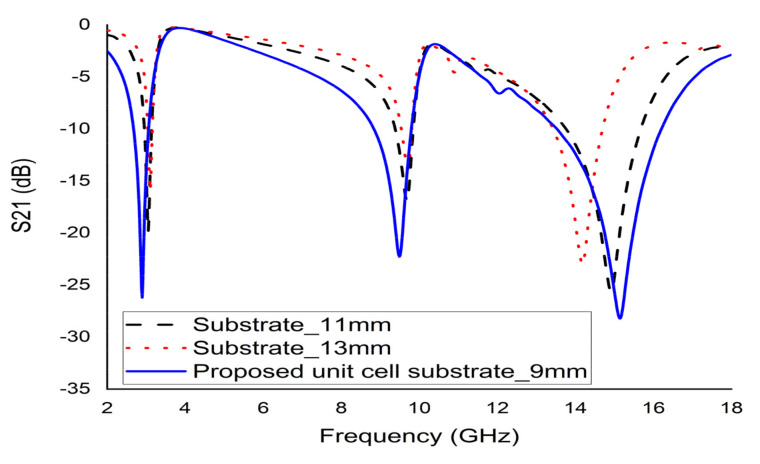
S21 for the different sized unit cell.

**Figure 5 micromachines-12-00878-f005:**
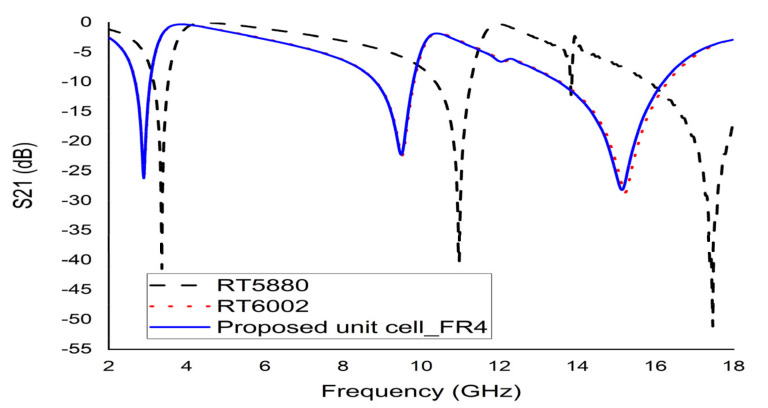
S21 for different substrate materials.

**Figure 6 micromachines-12-00878-f006:**
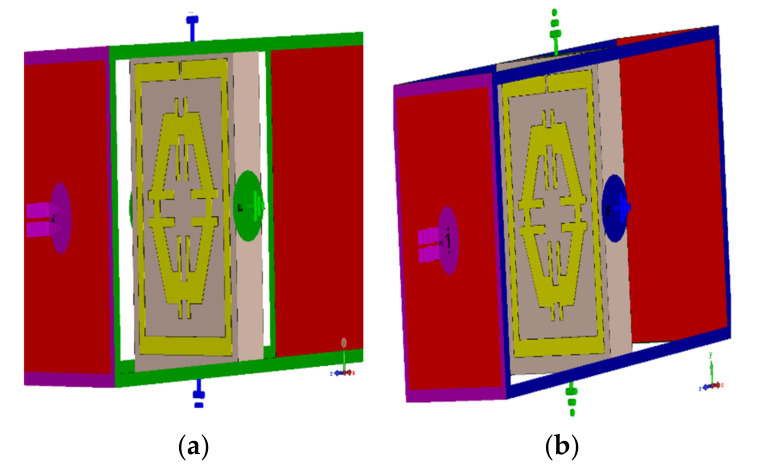
Simulation setup for changing field propagation, (**a**) magnetic field applied *x*-axis (**b**) magnetic field applied *y*-axis.

**Figure 7 micromachines-12-00878-f007:**
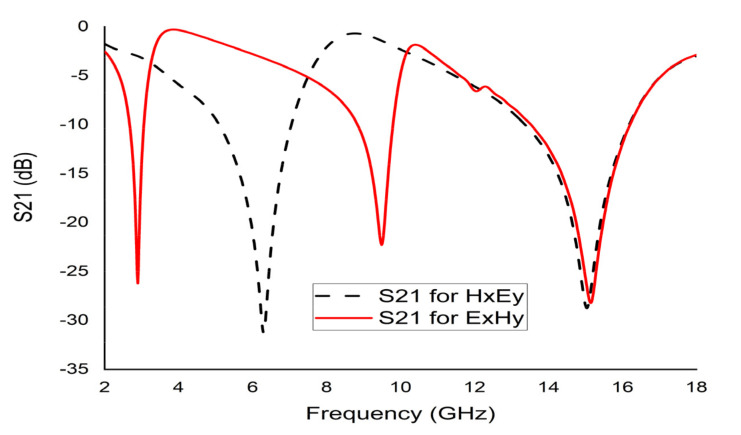
S21 for changing field direction.

**Figure 8 micromachines-12-00878-f008:**
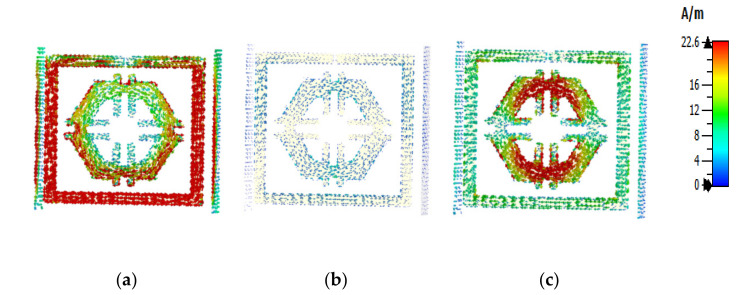
Surface current: (**a**) 2.89 GHz, (**b**) 9.42 GHz, and (**c**) 15.16 GHz.

**Figure 9 micromachines-12-00878-f009:**
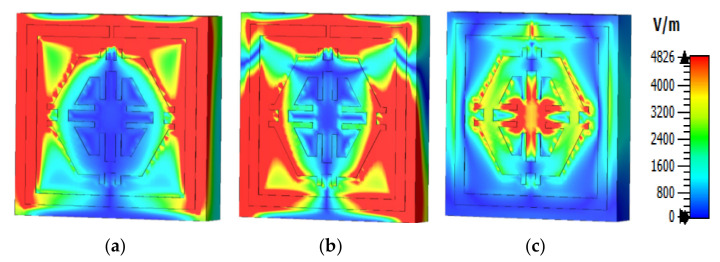
Electric field pattern: (**a**) 2.89 GHz, (**b**) 9.42 GHz, and (**c**) 15.16 GHz.

**Figure 10 micromachines-12-00878-f010:**
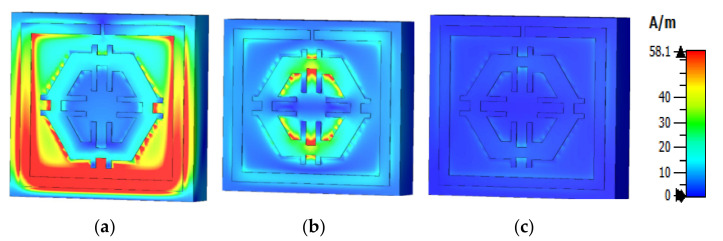
Magnetic field distribution (**a**) 2.89 GHz, (**b**) 9.42 GHz, and (**c**) 15.16 GHz.

**Figure 11 micromachines-12-00878-f011:**
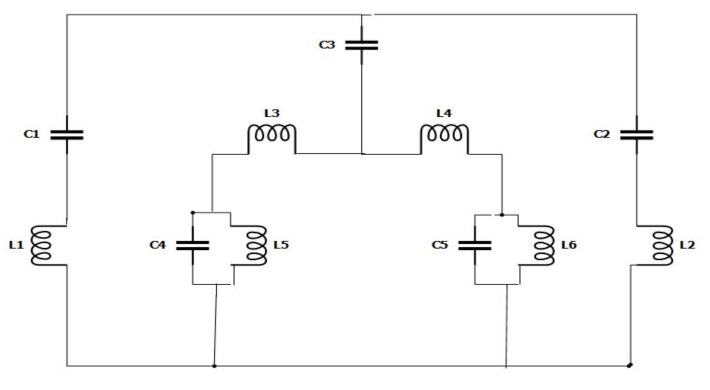
Equivalent circuit model of a unit cell.

**Figure 12 micromachines-12-00878-f012:**
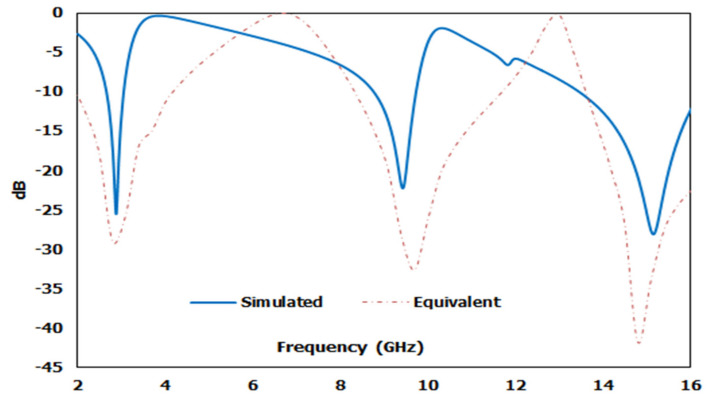
Comparison of CST and ADS transmission coefficient S21 results.

**Figure 13 micromachines-12-00878-f013:**
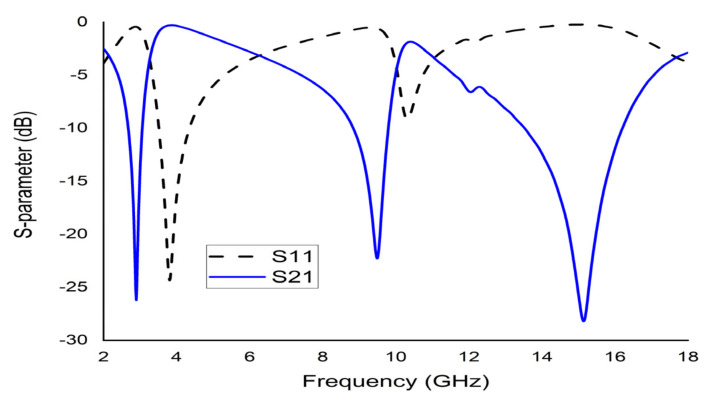
CST results of S11 and S21 of the unit cell.

**Figure 14 micromachines-12-00878-f014:**
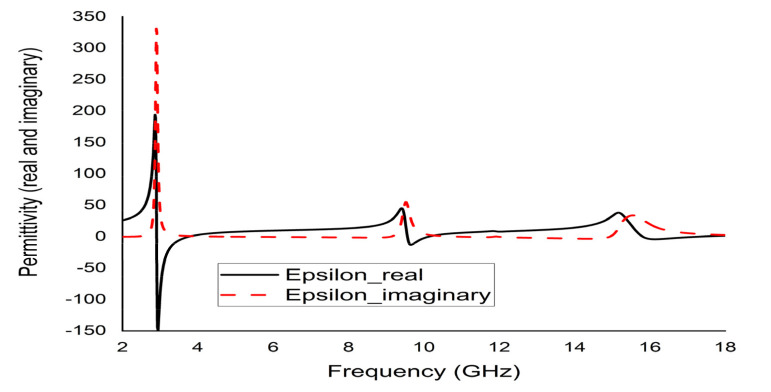
Real and imaginary permittivity (ε) of the unit cell.

**Figure 15 micromachines-12-00878-f015:**
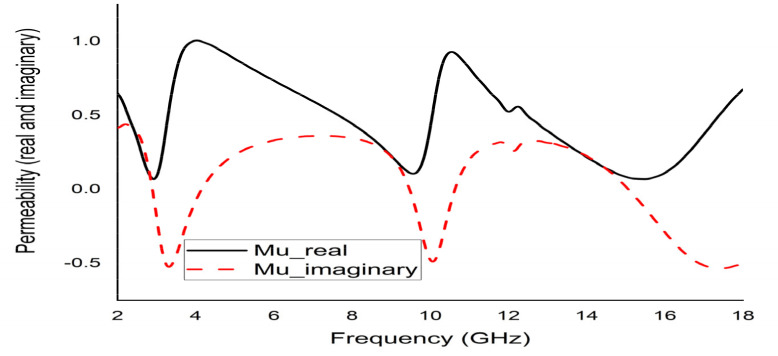
Real and imaginary permeability (µ) of the unit cell.

**Figure 16 micromachines-12-00878-f016:**
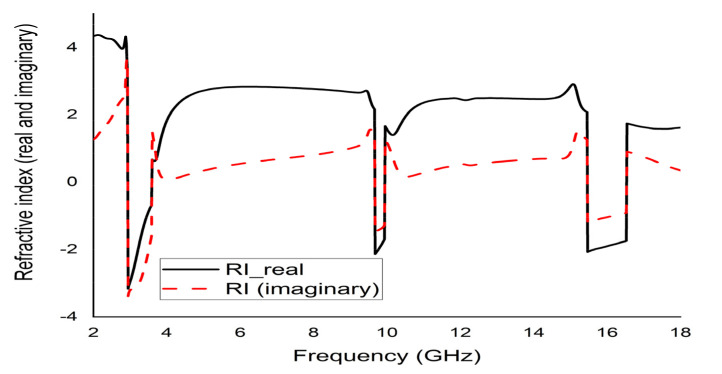
Real and imaginary refractive index (n) of the unit cell.

**Figure 17 micromachines-12-00878-f017:**
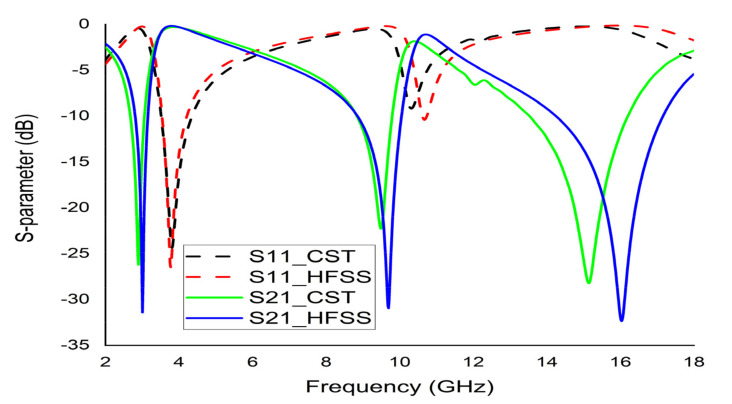
Validation of S11 and S21 of the unit cell based on CST and HFSS.

**Figure 18 micromachines-12-00878-f018:**
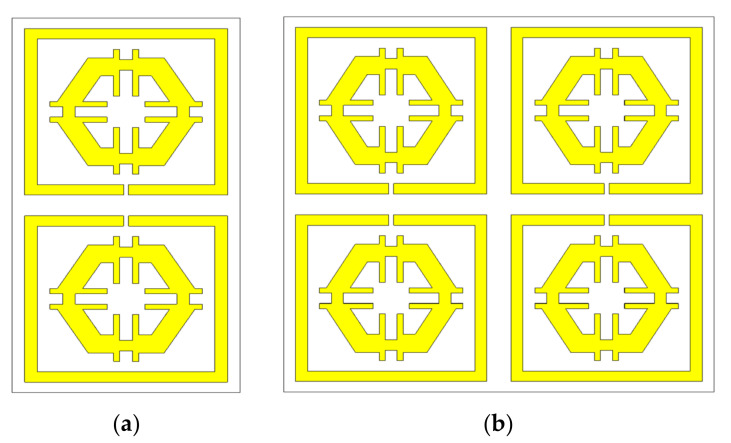
(**a**) 2 × 1 array, and (**b**) 2 × 2 array.

**Figure 19 micromachines-12-00878-f019:**
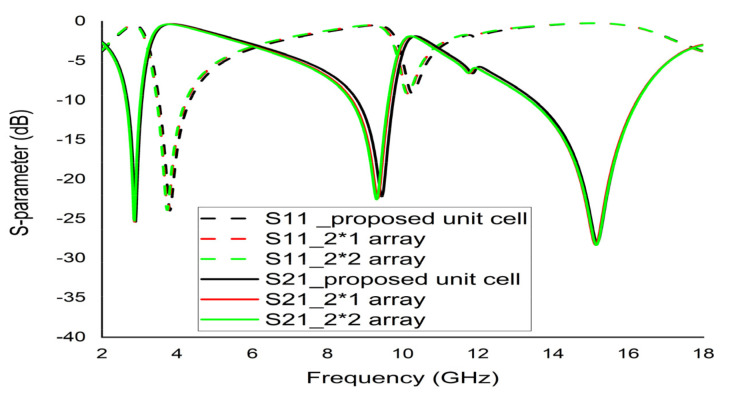
Comparison of S11 and S21 between the unit cell and array combinations.

**Table 1 micromachines-12-00878-t001:** Unit cell parameters.

Parameter	Dimension (mm)	Parameter Dimension (mm)
a	9	G 0.50
b	9	g 0.20
W1	8	W 0.50
W2	8	y 1

**Table 2 micromachines-12-00878-t002:** S21 for various designs.

Subdesign	Resonance Frequency (GHz)	Types Structure	Resonance Apex (dB)	Covering Bands
Design-1	2.9, 10	SSR	−27.44, −26.19	S, X
Design-2	3, 9.8, 17.33	SSR+ Hexagon	−25.76, −20.83, −28.28	S, X, Ku
Design-3	2.89, 10, 17.2	Modified hexagon	−27.4, −30, −29	S, X, Ku
Proposed design	2.89, 9.42, 15.16	Formatted hexagon	−25, −22.24,−28	S, X, Ku

**Table 3 micromachines-12-00878-t003:** Extracted properties of the unit cell.

Parameter	Frequency (GHz)	Extracted Property
Transmission coefficient, S21	2.89, 9.42, 15.16	S21 < −20 dB
Permittivity, ε_r_	2.9, 9.54,15.36	ε_r_ < 0
Permeability, µ_r_	2.86, 9.47, 15.60	µ_r_ ≈ 0
Refractive index, n	2.90, 9.60, 15.47	n < 0

**Table 4 micromachines-12-00878-t004:** Comparison of resonance frequencies among the unit cell and arrays.

Types of Simulation	Unit Cell	2 × 1 Array	2 × 2 Array	Max. Shift of Frequency (%)
1st resonance (GHz)	2.89	2.87	2.87	0.69
2nd resonance (GHz)	9.42	9.32	9.29	1.3
3rd resonance (GHz)	15.16	15.12	15.14	0.26

**Table 5 micromachines-12-00878-t005:** Authentication between the previous configurations and proposed configuration.

Author Name	Design Shape	Dimension (mm^3^)	Metamaterial Type	EMR (*λ*/L)
Mallik et.al.	U	25 × 25 × 1	LHM	1.99
Zhou Z et.al.	Modified H	9 × 9 × 1	Chiral	3.50
Islam et.al	H	30 × 30 × 1.6	LHM	3.65
Hossain et.al.	G	12 × 12 × 1.6	NIM	11.90
Faruque et.al.	SSR	10 × 10 × 1.6	LHM	9.10
Hossain et.al.	C	12 × 12 × 1.6	NM	7.44
Hasan et.al.	Modified G	10 × 10 × 1.6	SNM	3.98
Proposed metamaterial	Formatted hexagon	9 × 9 × 1.6	ENG	11.53

## Data Availability

All the data is available within the manuscript.

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
