# Peer review of "Modified Hexagonal Split Ring Resonator Based on an Epsilon-Negative Metamaterial for Triple-Band Satellite Communication"

_micromachines, 2021, doi:10.3390/mi12080878_

Round 1
Reviewer 1 Report
In this paper, the author described a new and novel metamaterial assembly whose unit cell consists of one split ring resonator with a modified hexagonal shaped metal stripe. Besides, the author performed different types of validation for the reliability of the performance of the proposed metamaterial. This projected metamaterial unit cell can be utilized for various applications including satellite communication, long-distance radio communication, etc. This work is highly recommended for publication in the micromachines journal in the present form.
Author Response
As attached.

Reviewer 2 Report
Authors have discussed a triple band epsilon negative (ENG) metamaterial on the basis of Split Ring Resonator (SSR) with a modified hexagonal shaped metal strip. I think, paper is interesting, I would propose changes as follows:
- Authors should clearly describe all the symbols in equations (1)-(4).
- It is not clear, if the equations (1), (2) serve as the homogenization tool for the system under study.
- English language should be improved as there are some errors such as Applying Drude function [24] ; , etc.
- Authors should stress novelty of their research as a wide spectrum of publications has already been dedicated to the modelling and design of the metamaterials and their properties.
- It would be highly desirable to validate the obtained results by providing some experimental investigations.
Author Response
As attached.

Round 2
Reviewer 2 Report
Accept.